# Cloning of Metalloproteinase 17 Genes from Oriental Giant Jellyfish *Nemopilema nomurai* (Scyphozoa: Rhizostomeae)

**DOI:** 10.3390/toxins14080519

**Published:** 2022-07-29

**Authors:** Du Hyeon Hwang, Yunwi Heo, Young Chul Kwon, Ramachandran Loganathan Mohan Prakash, Kyoungyeon Kim, Hyunju Oh, Ramin Seyedian, Al Munawir, Changkeun Kang, Euikyung Kim

**Affiliations:** 1Department of Pharmacology and Toxicology, College of Veterinary Medicine, Gyeongsang National University, Jinju 52828, Korea; pooh9922@hanmail.net (D.H.H.); unwi0510@gnu.ac.kr (Y.H.); skyward2@hanmail.net (Y.C.K.); mohanprakash111@gmail.com (R.L.M.P.); ckkang@gnu.ac.kr (C.K.); 2Ocean Climate & Ecology Research Division, National Institute of Fisheries Science, Busan 46083, Korea; weedy7411@korea.kr (K.K.); hyunjuoh@korea.kr (H.O.); 3Department of Pharmacology, Bushehr University of Medical Sciences, Bushehr 14174, Iran; raminseyedian@gmail.com; 4Pathology Laboratory, Medical Faculty, University of Jember, Jember 68126, Indonesia; almunawir.fk@unej.ac.id

**Keywords:** jellyfish venom, metalloproteinase, sequencing

## Abstract

We previously demonstrated that *Nemopilema nomurai* jellyfish venom metalloproteinases (JVMPs) play a key role in the toxicities induced by *N. nomurai* venom (NnV), including dermotoxicity, cytotoxicity, and lethality. In this study, we identified two full-length JVMP cDNA and genomic DNA sequences: JVMP17-1 and JVMP17-2. The full-length cDNA of JVMP17-1 and 17-2 contains 1614 and 1578 nucleotides (nt) that encode 536 and 525 amino acids, respectively. Putative peptidoglycan (PG) binding, zinc-dependent metalloproteinase, and hemopexin domains were identified. BLAST analysis of JVMP17-1 showed 42, 41, 37, and 37% identity with *Hydra vulgaris*, *Acropora digitifera*, *Megachile rotundata,* and *Apis mellifera* venom metalloproteinases, respectively. JVMP17-2 shared 38 and 36% identity with *H. vulgaris* and *A*. *digitifera*, respectively. Alignment results of JVMP17-1 and 17-2 with other metalloproteinases suggest that the PG domain, the tissue inhibitor of metalloproteinase (TIMP)-binding surfaces, active sites, and metal (ion)-binding sites are highly conserved. The present study reports the gene cloning of metalloproteinase enzymes from jellyfish species for the first time. We hope these results can expand our knowledge of metalloproteinase components and their roles in the pathogenesis of jellyfish envenomation.

## 1. Introduction

In general, the venom of poisonous animals is a cocktail of bioactive proteins, peptides, and small molecules that incapacitate or digest their prey. The hundreds of proteins found in venom include toxins, nontoxic proteins, and many different enzymes. In particular, metalloproteinases, which are are highly abundant in snake venoms [1,2,3] and jellyfish venoms [4,5], are involved in venom-associated pathogenesis. The toxicological effects of snake venom metalloproteinases (SVMPs) include local and systemic hemorrhages [6], anti-coagulation [7], and inflammation and necrosis [8]. SVMPs are typically organized into three main groups (P-I to P-III) based on their domain organization [8,9]. Class P-I is the smallest SVMP, with only a metalloproteinase (M) domain, P-II contains an M domain and a disintegrin (D) domain, and P-III includes M, D, and cysteine-rich (C) domains. P-III SVMPs are further divided into subclasses based on their post-translational modifications, such as dimerization (P-IIIc) or proteolysis between the M and D domains (P-IIIb). P-IIId SVMP contains an additional C-type lectin-like domain [10].

Over the last few decades, *Nemopilema nomurai* jellyfish (Phylum Cnidaria) have bloomed in the seas of Korea, China, and Japan, causing substantial damage to the fishing industry as well as causing numerous jellyfish stings to sea bathers. Jellyfish stings can cause redness, local edema, a burning sensation, and vesicular eruption [11]. In addition, venom can infrequently cause systemic reactions such as hyperventilation, shock, paralysis, cardiovascular collapse, and even death [12]. Recently, we have suggested that *N. nomurai* jellyfish venom metalloproteinases (JVMPs) play a key role in the toxicities induced by *N. nomurai* venom (NnV), including dermotoxicity, cytotoxicity, and lethality [4,13,14,15]. Similar to snake venom, jellyfish venom also contains phospholipases [16,17,18]. We have previously reported that NnV demonstrates phospholipase, hyaluronidase, and metalloproteinase activity and that it can induce cytotoxicity in numerous cell types [4]. Here, we report the identification of the full-length cDNA and gene structures of two isoforms of JVMPs from the scyphozoan jellyfish *N. nomurai* for the first time.

## 2. Results

### 2.1. Metalloproteinases Components of NnV

The proteolytic activity of NnV was investigated by zymography analyses using gelatin as the substrate. Figure 1A shows the gelatinolytic activity at various molecular masses, with particularly strong signals at approximately 25 kDa and in the range of 50–70 kDa. Most of the gelatinolytic activities of NnV were significantly abolished by treatment with conventional metal chelators, suggesting that this was caused by metalloproteinases.

### 2.2. Full-Length cDNA Sequence Analysis of N. nomurai JVMP17-1 and 17-2

The full-length the cDNAs of the *N. nomurai* JVMP isoforms JVMP17-1 and JVMP17-2 were amplified with an oligo(dT)_18_ primer and primers based on the partial transcript sequences. The full-length cDNAs of the JVMP17-1 and 17-2 isoforms contain 1614 (GenBank accession no. MW727214) and 1578 nucleotides (nt) (GenBank accession no. MW727215), respectively, and the deduced amino acid (aa) sequences encode 537 aa and 525 aa, respectively (Figure 2). Cleavage of the putative signal peptide of JVMP17-1 occurs between aa 24 and 25 and between aa 18 and 19 in JVMP17-2. BLAST analysis of the JVMP17-1 gene sequence showed 42, 41, 37, and 37% identity with H. vulgaris, A. digitifera, M. rotundata, and A. mellifera venom metalloproteinases, respectively. JVMP17-2 showed 38 and 36% identity with H. vulgaris and A. digitifera, respectively (data not shown). Alignment analysis of these two genes revealed that they have PG-binding, zinc-dependent metalloproteinase, and hemopexin domains. Furthermore, nine TIMP-binding surfaces, metal-binding sites, and active sites were found to be highly conserved. The zinc-binding motif (HExxHxxxxxH) sequence was also highly conserved (Figure 2).

### 2.3. Genomic DNA Sequences of N. nomurai JVMP17-1 and 17-2

To determine the gene sequences of JVMP17-1 and JVMP17-2, PCR was performed using specific primers designed using the full-length cDNA sequences. The PCR products were cloned into the pGEM-T Easy Vector and the clones were confirmed by EcoRI digestion. The whole genome sequences of JVMP17-1 and JVMP17-2 comprised 5687 (Figure 3A) and 6661 base pairs (bp) (Figure 3B), respectively. The JVMP17-1 gene contained seven distinct exons, while JVMP17-2 only contained six distinct exons. The classical 5′ donor (GT) and 3′ acceptor (AG) splice sites were present at each exon/intron boundary (Figure 4A,B).

## 3. Discussion

To identify *N. nomurai* JVMP17-1 and 17-2 genes, we synthesized the first-strand cDNA and used it to perform the rapid amplification of cDNA ends polymerase chain reaction (RACE PCR). The complete open reading frame (ORF) of JVMP17-1 contained 1614 nucleotides, including a stop codon, and encoded 537 amino acids (Figure 4A). BLAST analysis of the cDNA showed that it shared 42, 41, 37, and 37% identity with *H. vulgaris*, *A*. *digitifera*, *M*. *rotundata,* and *A. mellifera* venom metalloproteinases, respectively. According to the SignalP 4.1 program, a putative signal peptide exists between amino acids 24 and 25. The primary structure of the protein predicted using InterProScan has PG-binding, zinc-dependent metalloproteinases, and hemopexin domains. Furthermore, nine TIMP-binding surfaces, metal-binding sites, and active sites were found to be conserved. The zinc-binding motif (HExxHxxxxxH) sequence was also highly conserved (Figure 2).

SVMPs are relatively well defined [19]. They have domain structures composed of signal peptide, propeptide, zinc-dependent metalloproteinase (M), disintegrin (D), and cysteine-rich (C) domains. Whereas, JVMP17-1 and 17-2 have signal peptide, PG-binding, M, and hemopexin domains (Figure 5). There is very low homology between the M domain of JVMP17-1 and 17-2 and the SVMP M domain sequences (GenBank accession number: O42138 of *Agkistrodon contortrix laticinctus*, C9E1S0 of *Agkistrodon piscivorus leucostoma*, Q9w6M5 of *Deinagkistrodono acutus*, Q8AWI5 of *Gloydius halys*, ABG26979 of *Sistrurus catenatus edwardsi*, A8QL59 of *Naja atra*, O93523 of *Bothrops jararaca*, Q8QG88 of *Bothrops insularis*, Q90ZI3 of *Protobothrops flavoviridis*, ABN72537 of *Bungarus multicinctus*, C9E1R8 of *Crotalus viridis*, and Q9DGB9 of *Crotalus atrox*) (data not shown). In general, SVMPs have a propeptide domain between the signal peptide and M domains that causes the activation of the SVMP when hydrolyzed. In contrast, JVMP17-1 has a PG-binding domain. In many enzymes, the PG-binding domain usually plays a role in binding to the peptidoglycan in bacterial cell walls, inducing their degradation [20,21,22]. Eukaryotic matrix metalloproteinases (MMPs) have a similar PG-binding domain that can catalyze extracellular matrix degradation in association with arthritis [23], tumor invasion [24], and immune defense mechanisms [25]. The disintegrin domain of SVMP binds and inhibits the integrin of the platelets or endothelial cell membrane in blood vessels. However, the disintegrin of our novel JVMPs has a hemopexin domain (KGS and KGD motif), which is a heme-binding moiety that plays an important role in cell migration [26] or in heme transfer to the liver for the inhibition of oxidative stress [27] (Figure 5).

The genome structures of JVMP17-1 and JVMP17-2 were determined using specific PCR primers designed using the full-length cDNA sequences. JVMP17-1 contains 5687 bp (Figure 3A) and has seven distinct exons. Interestingly, the dinucleotide sequences at the 5′ donor (GT) and 3′ acceptor (AG) splice sites in the introns were highly conserved (Figure 4A). JVMP17-2 contains 6661 bp (Figure 3B) and has six distinct exons. The 5′ donor and 3′ acceptor splice sites of JVMP17-2 were also highly conserved (Figure 4B).

## 4. Conclusions

In the present study, we identified the full-length cDNAs and gene sequences of the novel proteins JVMP17-1 and JVMP17-2 from the scyphozoan jellyfish *N. nomurai*. To determine the functions of these two enzymes in the future, it will be necessary to isolate them from NnV or generate recombinant proteins using in vitro expression systems in *Escherichia coli* or yeast. To the best of our knowledge, this study is the first to report the full-length cDNAs, gene sequences, and the primary protein structures of two metalloproteinase isoforms from jellyfish species.

## 5. Materials and Methods

### 5.1. Jellyfish Collection and Nematocyst Preparation

*N. nomurai* jellyfish specimens were captured near the coast of the Republic of Korea and were immediately transferred to the laboratory on ice. Nematocysts were isolated using a previously described method [28] with slight modifications. The dissected tentacles were washed several times with ice-cold seawater to remove any debris. They were then placed in three volumes (*v*/*v*) of cold seawater for 24 h at 4 °C with gentle swirling. After autolysis, the supernatant was centrifuged at 1000× *g* for 5 min to harvest the nematocysts. The autolysis process for the tentacles was repeated for an additional 3–4 days, and seawater was replaced daily. The harvested nematocyst pellets were stored in a deep freezer and dried using a freeze dryer (lyophilizer), and the final powder was stored at −70 °C until required.

### 5.2. Venom Preparation

Freeze-dried nematocyst powder was used to extract jellyfish venom using a previously described method [5] with a minor amendment. Briefly, venom was extracted from 70 mg of nematocyst powder using glass beads (approximately 8000 beads; 0.5 mm in diameter) in 1 mL of cold phosphate-buffered saline (PBS, pH 7.4). The mixtures were vortexed for 30 s, and this step was repeated five times with intermittent cooling on ice. The venom extracts were then transferred to new microfuge tubes and centrifuged (22,000× *g*) at 4 °C for 30 min. The supernatant was used as the NnV in the present study. The Bradford assay (Bio-Rad, Hercules, CA, USA) was used to determine the protein concentration in the venom [29].

### 5.3. Metalloproteinase Analysis of NnV

The metalloproteinase activity of NnV was analyzed by a proteolytic zymography assay using gelatin as the substrate, as previously described [30]. To prepare the zymography gels, gelatin (2 mg/mL) and thrombin (0.01 U/mL) dissolved in 20 mM sodium phosphate buffer (pH 7.4) were copolymerized with 12% polyacrylamide. NnV (5 µg) was loaded in a non-reducing sample buffer before electrophoresis at 15 mA/gel at 4 °C. The SDS was removed by washing the gel twice in 2.5% Triton X-100 for 20 min. The gel was then incubated in 20 mM Tris (pH 7.4) and in 0.5 mM calcium chloride at 37 °C for 16 h before staining with 0.125% Coomassie blue. Where required, the metalloproteinase protease inhibitors 1,10-phenanthroline, tetracycline, and EDTA were added to the incubation buffer of the appropriate gel at a final concentration of 10 mM. The zymography assay was performed as previously described. Clear zones in the gel indicate regions with proteolytic activity.

### 5.4. Total RNA Extraction

Total RNA was extracted using a previously described method [31]. Briefly, lyophilized tentacle powder was dissolved in lysis buffer, and total RNA was purified by ethanol precipitation. The pellet was dissolved in diethyl pyrocarbonate-treated, nuclease-free water and treated with DNase I (NEB, Ipswich, MA, USA). The total RNA was used as a template for RACE after heat treatment at 75 °C for 10 min to inactivate the DNase I.

### 5.5. Rapid Amplification of cDNA Ends (RACE)

The 3′-RACE PCRs used for the two genes were performed with specific forward primers (JVMP17-1: 5′-GATGGAGGACAGCAGACGAATGGC-3′; JVMP17-2: 5′-GGATACCCAAGGAGCGTTTGGGAG-3′) designed based on the transcriptome sequence data and an oligo (dT)_18_ primer. The 5′-RACE PCRs were performed using the SMARTer RACE cDNA Amplification Kit (Clontech, Mountain View, CA, USA) according to the manufacturer’s instructions. First-strand cDNA for 5′-RACE was synthesized from total RNA with gene-specific primers (JVMP17-1: 5′-GTCCATCGTATCGGCCGTGACATC-3′; JVMP17-2: 5′-GCGAGGTAGTTAAGTCCTTCCTGG-3′) and the SMARTer II A oligonucleotide. The 5′-RACE PCR was performed using an Advantage^®^ 2 PCR Enzyme Kit (Clontech, Mountain View, CA, USA). All PCR products were purified using an Expin™ PCR SV purification kit (cat. no. 103-102; GeneAll Biotechnology Co., Ltd., Seoul, Republic of Korea), cloned into the pGEM-T^®^ Easy Vector System (Promega, Madison, WI, USA), and sequenced using an ABI PRISM 3739 XL Genetic Analyzer (Applied Biosystems, Waltham, MA, USA).

### 5.6. Genomic DNA Sequences of N. nomurai JVMP17-1 and JVMP17-2

Genomic DNA was purified from the whole body of the jellyfish using the cetyltrimethyl ammonium bromide method [31]. Specific primers for the JVMP17-1 and JVMP17-2 genomic DNA sequences were designed based on their full-length cDNA sequences.

### 5.7. Nucleotide Sequence Analysis

A homology search of full-length cDNA sequences was performed using the NCBI BLAST program (http://www.ncbi.nlm.nih.gov/BLAST/, accessed date (10 March 2021)). Protein domains were predicted using the InterProScan search tool (www.ebi.ac.uk/Tools/InterProScan/, accessed date (5 June 2022)). The signal peptide cleavage site in the deduced amino acid sequences was predicted using SignalP 4.1 (http://www.cbs.dtu.dk/services/SignalP, accessed date (5 June 2022)). The sequence identity values for the deduced amino acid sequences are available on the EMBL-EBI website (http://www.ebi.ac.uk/Tools/emboss/align/, accessed date (5 June 2022)).

All data used for statistical material can be found in the Appendix A.

## Figures and Tables

**Figure 1 toxins-14-00519-f001:**
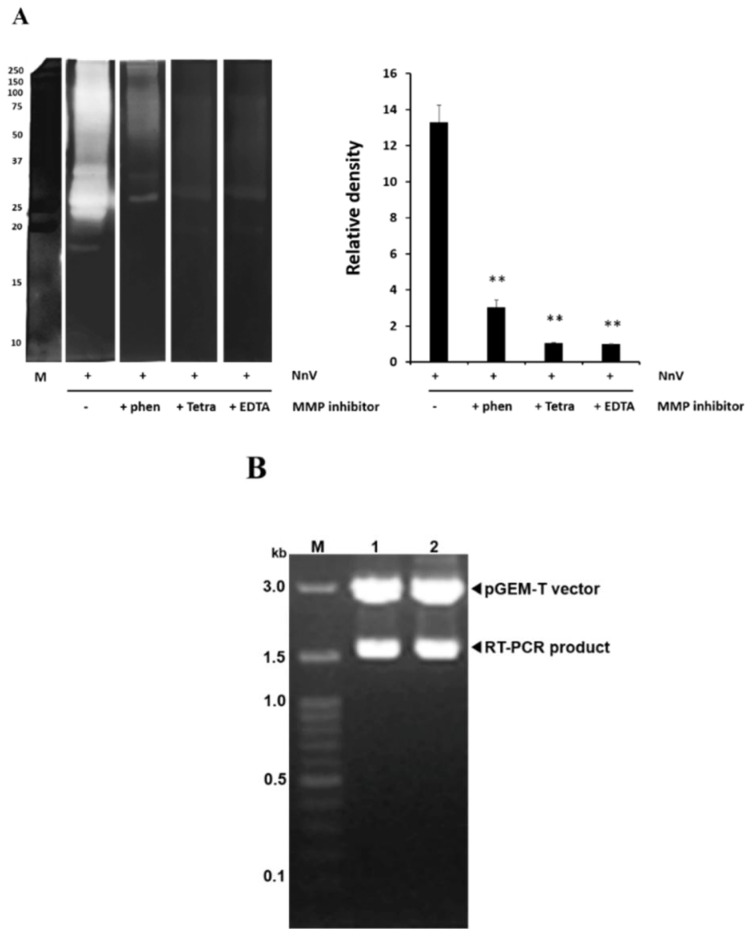
JVMP metalloproteinase activity (**A**) and JVMP 17-1 and 17-2 cDNA clones (**B**). (**A**) Analysis of the metalloproteinase activity in *N. nomurai* jellyfish venom (NnV) using gelatin zymography assays performed in the presence or absence of several metalloproteinase inhibitors (1,10-phenanthroline, tetracycline, and EDTA, 10 mM). Quantification of the metalloproteinase activity. Data represent mean ± SD of the three fields. ** *p* < 0.01 compared to the NnV group. (**B**) Agarose gel electrophoresis of the pGEM-T/JVMP cDNA after EcoRI digestion. M: 100-bp size marker; lane 1: pGEM-T/JVMP17-1 cDNA; lane 2: pGEM-T/JVMP17-2 cDNA.

**Figure 2 toxins-14-00519-f002:**
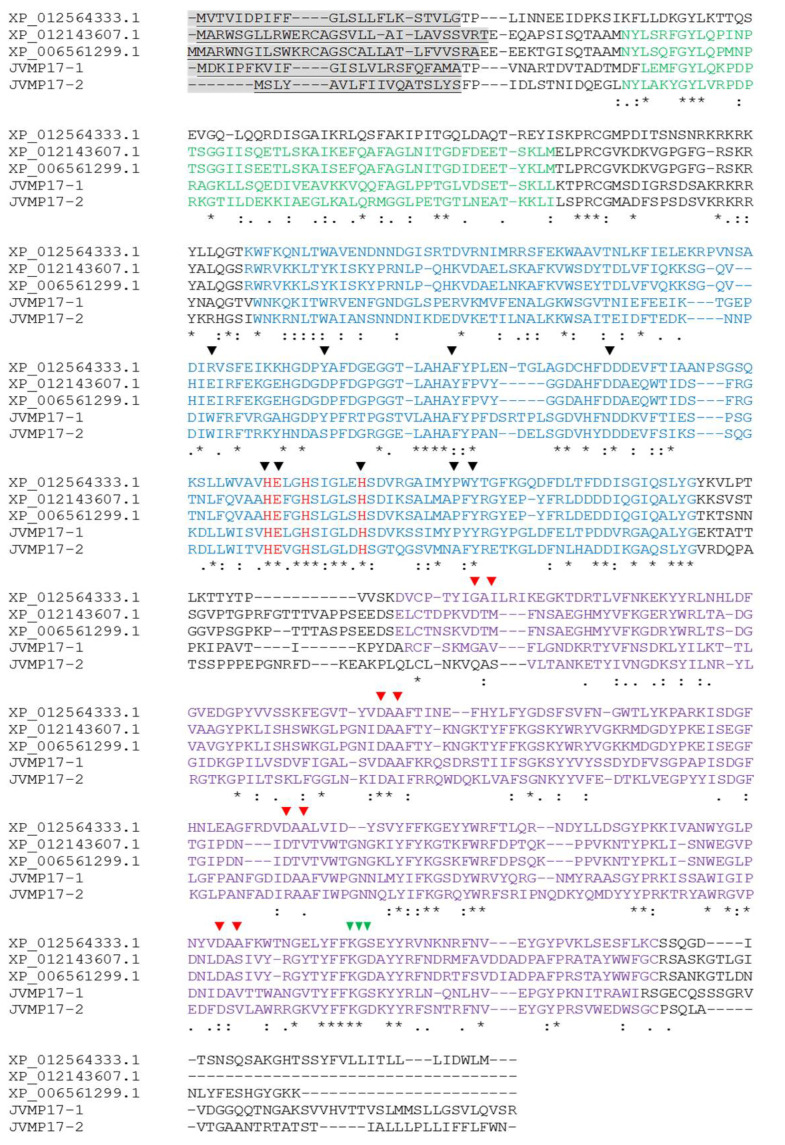
Alignment analysis of JVMP17-1 and 17-2 compared to that of other metalloproteinases. XP_012564333.1: Hydra vulgaris; XP_012143607.1: Megachile rotundata; XP_006561299.1: Apis mellifera; shaded letters: signal peptides; green letters: PG binding surface; red letters: active site of metalloproteinase; blue letters: zinc-dependent metalloproteinase domain; (▼): TIMP-binding surfaces; (▼): metal (ion)-binding sites; (▼): disintegrin motif. Identical, similar, and weakly similar amino acids are indicated by asterisks, colons, and dots, respectively. Gaps are indicated by dashes.

**Figure 3 toxins-14-00519-f003:**
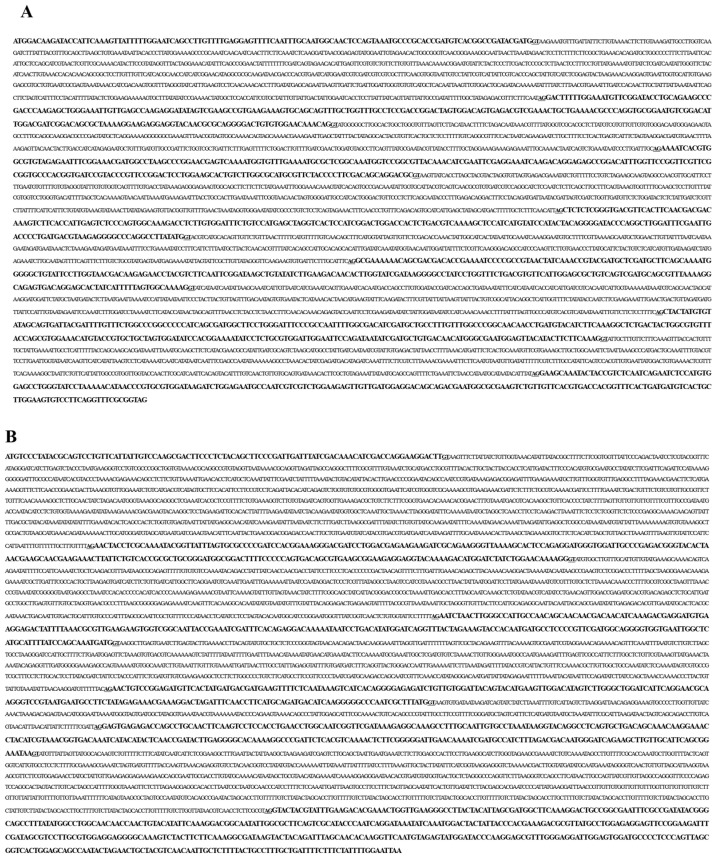
Whole genome sequences of JVMP17-1 (**A**) and JVMP17-2 (**B**).

**Figure 4 toxins-14-00519-f004:**
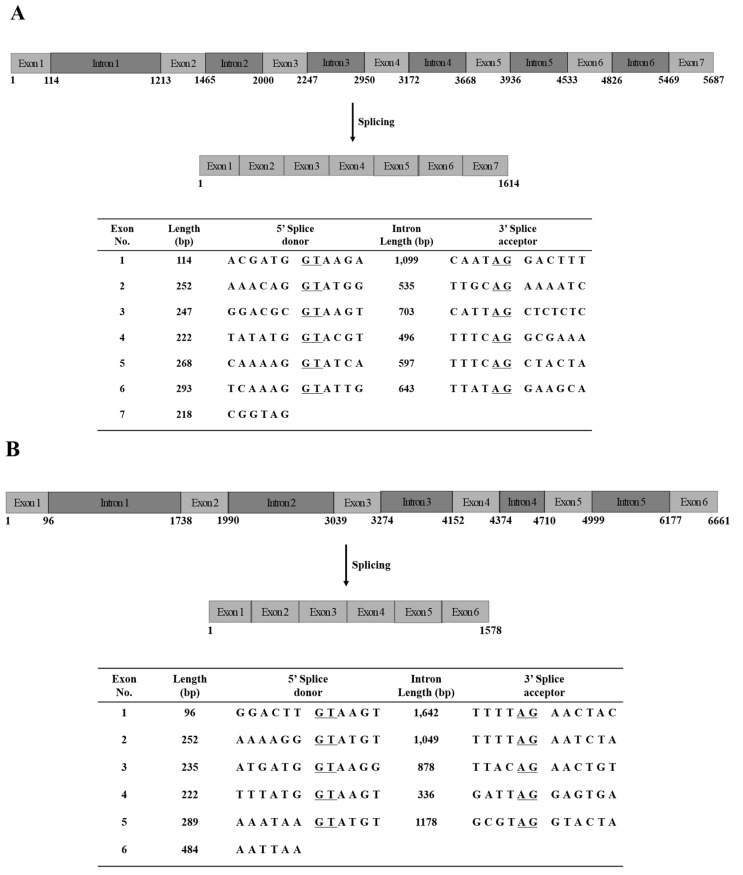
Organization of N. nomurai JVMP17-1 (**A**) and JVMP17-2 (**B**) genes. The 5′ acceptor and 3′ donor splice sites are underlined.

**Figure 5 toxins-14-00519-f005:**
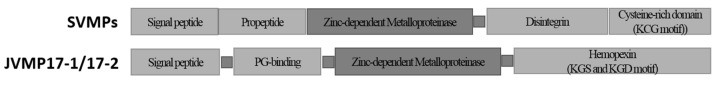
Comparison between SVMPs and *N. nomurai* JVMP17-1 and 17-2 proteins. Schematic representation of JVMP17-1 and 17-2 protein domain structure exhibits the conserved domains. SVMP shows domain structures composed of signal peptide, propeptide, zinc-dependent metalloproteinase (M), disintegrin (D), cysteine-rich, and hemopexin domains.

## Data Availability

Not applicable.

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
