# Peer review of "Cloning of Metalloproteinase 17 Genes from Oriental Giant Jellyfish Nemopilema nomurai (Scyphozoa: Rhizostomeae)"

_toxins, 2022, doi:10.3390/toxins14080519_

Round 1

Reviewer 1 Report

The manuscript entitled “Cloning of metalloproteinase 17 genes from oriental giant jelly-fish Nemopilema nomurai (Scyphozoa: Rhizostomeae)” aimed to identify two full-length JVMP cDNA and genomic DNA sequences, named JVMP17-1 and JVMP17-2 in addition to identify peptidoglycan binding, zinc-dependent metalloproteinase, and hemopexin domains. Authors indicated that these findings might expand current knowledge on metalloproteinase components and their roles in the pathogenesis of jellyfish envenomation. Based on its content this is a very well conducted and written article and I have only few remarks of the technical nature.

Could authors discus their findings in the possible therapeutic context of the venom derived from jellyfish.

Amreen Nisa et al (2021) Jellyfish venom proteins and their pharmacological potentials: A review. Int J Biol Macromol. 176 :424-436. doi: 10.1016/j.ijbiomac.2021.02.074.

Von Reumont BM et al (2022) Modern venomics-Current insights, novel methods, and future perspectives in biological and applied animal venom research. Gigascience. doi: 10.1093/gigascience/giac048.

The Figures could be in better resolution, especially Figure 3.

Authors are stating that data are not shown for the very low homology between the M domain of JVMP17-1 or 17-2 and SVMP M domain sequences. Why not report these data as Supplementary material?

Author Response

Answer 1:

Thank you for the valuable suggestion. We have discussed the therapeutic aspect of jellyfish venom in Introduction part and attached the related literature in Reference part of the revised manuscript according to the reviewer’s comment.

Answer 2:

We appreciate the precious comment. We have provided Figure 3 with bigger size images in the revised manuscript according to the reviewer’s comment.

Reviewer 2 Report

In this study, the authors have identified full-length genomic DNA sequences of jellyfish venom metalloproteinases - 17-1 and 17-2. The authors have done a good job in identifying the whole-genome sequence as well as the amino acid sequence. The alignment and BLAST analysis have revealed similarities to various other toxins. The data presented in this paper is good, however, the authors can elaborate on the Discussion section by comparing the similarities mentioned in the paper with various other toxins. An additional brief paragraph on the potential mechanism of the jellyfish toxin based on the knowledge of other compared toxins can be added as well. It would really be interesting to know the functions and mechanisms of these enzymes in in vitro studies.

Author Response

Answer1:

Thank you for the helpful suggestion. We have added the information in Supplementary Firure 1 of the revised manuscript according to the reviewer’s comment.

Answer2:

We appreciate the important suggestion. We have compared the sequence similarities with the MMPs of other species and provided the information in Supplementary table1 of the revised manuscript.
